# Evaluation of the practicability of a finger-stick whole-blood SARS-Cov-2 self-test adapted for the general population

**Thierry Prazuck**[1]*, **Jean Phan Van**[2], **Florence Sinturel**[3], **Frederique Levray**[3], **Allan Elie**[3], **Denise Camera**[4], **Gilles Pialoux**[4]

1 Department of infectious and tropical diseases, CHR Orléans, Orléans, France, 2 EDF Nuclear Plant, Saint-Laurent-des-Eaux, France, 3 EDF Nuclear Plant, Dampierre-en-Burly, France, 4 Department of infectious diseases, Hôpital Tenon, Assistance Publique des Hôpitaux de Paris, Sorbonne Université, Paris, France

* thierry.prazuck@chr-orleans.fr

**Data Availability Statement:** All relevant data are within the paper and its Supporting information file.

**Funding:** This study was funded by AAZ-LMB who provided the Presto rapid test but did not have a

## Abstract

### Background

COVID-19 (COronaVIrus Disease 2019) is an infectious respiratory disease caused by the novel SARS-CoV-2 virus. Point of Care (POC) tests have been developed to detect specific antibodies, IgG and IgM, to SARS-CoV-2 virus in human whole blood. They need to be easily usable by the general population in order to alleviate the lockdown that many countries have initiated in response to the growing COVID-19 pandemic. A real-life study has been conducted in order to evaluate the performance of the COVID-PRESTO® POC test and the results were recently published. Even if this test showed very high sensitivity and specificity in a laboratory setting when used by trained professionals, it needs to be further evaluated for practicability when used by the general public in order to be approved by health authorities for in-home use.

### Methods

143 participants were recruited between March 2020 and April 2020 among non-medical populations in central France (nuclear plant workers, individuals attending the Orleans University Hospital vaccination clinic and Orleans University Hospital non-medical staff). Instructions for use, with or without a tutorial video, were made available to the volunteers. Two separate objectives were pursued: evaluation of the capability of participants to obtain an interpretable result, and evaluation of the users' ability to read the results.

### Results

88.4% of the test users judged the instructions for use leaflet to be clear and understandable. 99.3% of the users obtained a valid result and, according to the supervisors, 92.7% of the tests were properly performed by the users. Overall, 95% of the users gave positive feedback on the COVID PRESTO® as a potential self-test. Neither age nor education had an influence.

role in study design, data collection and analysis, decision to publish, or preparation of the manuscript. Électricité de France (EDF) provided support for this study in the form of salaries for authors [JPV, FS, FL, AE, DC] but did not have any additional role in study design, data collection and analysis, decision to publish, or preparation of the manuscript. The specific roles of these authors are articulated in the 'author contributions' section. No additional external funding was received for this study.

**Competing interests:** The authors have read the journal's policy and have the following competing interests: JPV, FS, FL, AE, DC are paid employees of Électricité de France (EDF). There are no patents, products in development or marketed products associated with this research to declare. This does not alter our adherence to PLOS ONE policies on sharing data and materials.

## Conclusion

COVID-PRESTO® was successfully used by an overwhelming majority of participants and its use was judged very satisfactory, therefore showing promising potential as a self-test to be used by the general population. This POC test can become an easy-to-use tool to help detect whether individuals are protected or not, particularly in the context of a second wave or a mass vaccination program.

## Introduction

In Wuhan, China, the end of the year 2019 marked the emergence of a new type of pneumonia caused by a then-unknown agent. Shortly after the first reports of the disease, the Chinese health authorities and the World Health Organization announced that a newly-discovered type of coronavirus was responsible for the disease. This new virus was named SARS-CoV-2 (Sever Acute Respiratory Syndrome-CoronaVirus-2). This virus is a new member of the coronavirus family that already includes SARS-CoV and MERS-CoV responsible for the SARS outbreak in 2003, and an ongoing outbreak that started in 2012 in the Middle East, respectively.

The disease caused by the SARS-CoV-2 virus is called COVID-19 (COronaVIrus Disease-2019). The site of infection is located on the upper/lower respiratory tract [1]. The mean incubation period is approximatively 5.2 days and the most common symptoms are dry cough, fever and fatigue. Other symptoms include anosmia (loss of smell), ageusia (loss of taste), headache, sore throat and in the most severe cases, acute respiratory distress syndrome.

Within the last 6 months, according to the Johns Hopkins University Coronavirus resource center, the COVID-19 pandemic has spread over 188 countries; leaving very few regions of the world untouched. At the end of January 2020, the WHO declared this outbreak to be a global health emergency. The mark of 100,000 deaths was reached on April 12th, leading to a little less than 350,000 deaths on May 26th for a total of 5 519,878 confirmed cases. Governments have taken extreme measures to try to slow the spreading of the virus by imposing strict social distancing rules and it is estimated that 1.7 billion people have been confined at home worldwide.

In most countries, these measures have been effective in slowing down virus transmission. In May 2020, several governments started easing the confinement rules but questions remain with regards to testing, controlling and tracking every case in a large population at least until an effective treatment or vaccine can be found. To this end, the development of new testing tools that could be distributed to the population on a large scale is crucial. Serological tests able to detect the specific antibodies against the SARS-CoV-2 in people seem to be particularly good candidates because they are fairly easy to use without extensive training.

An extensive list of tests available worldwide can be found on the FIND foundation website [2]. As of October 10th 2020, around 50 COVID-19 serological tests are commercially available in the US through the Emergency Use Authorization (EUA) which has been granted to the test manufacturers by the Food and Drug Administration (FDA) [2]. The FDA has provided guidance for manufacturers of serological tests in order to promote and facilitate rapid market access on the basis that such tests could help provide crucial information about the prevalence of COVID-19 infections in different communities [3]. However, an NIH independent evaluation has shown that a concerning number of commercial serological tests are not being appropriately promoted or show poor performance [3]. In Europe, according to regulations, manufacturers should submit data regarding "handling suitability of the device in view of its intended purpose for self-testing" for assessment by notified bodies before making it available to the public [4]. The European Centre for Disease Prevention and Control reports that several

COVID-19 Rapid Diagnostic Tests (RDTs) or Point of Care tests (POCs) are being marketed with incomplete and even sometimes fraudulent documentation and unsubstantiated claims, some of these tests being sold as self-tests. Consequently, several European countries have banned the marketing of such tests until further notice [5]. There is, therefore a crucial need for more documented studies regarding those tests [6, 7].

From a regulatory standpoint, "self-testing" is a more stringent regulatory category of in vitro diagnosis tests compared to the regular kind, intended for professional use only. The French health authority (Haute Autorité de Santé, HAS) estimates that, in absence of reliable data regarding the available self-tests, it is premature to promote their use [8]. COVID-PRESTO® aims to be one of the first SARS-CoV-2 serological tests to be officially approved as a reliable and accurate self-test in order to offer a viable option for large-scale testing. It is important to bear in mind that these tests are not designed to detect an ongoing infection but rather a prior infection which would provide intelligence about the true prevalence of the virus. However, a positive test will not mean that the tested individual is no longer infectious but will give the information that he/she was infected at some point in the past. Currently, there is no solid evidence that a prior infection will offer strong immunity towards a second exposure and if so for how long [9]. Nevertheless, a pre-published study demonstrated that most of the patients having presented a mild form of the COVID-19, developed neutralizing antibodies [10].

The AAZ COVID-PRESTO® is an easy-to-use Point of Care (POC) test, device based on a lateral flow chromatographic immunoassay from a single drop of blood (Fig 1). Briefly, a lancet needle included in the kit is used to draw a drop of blood from the fingertip. The blood sample is then collected with a capillary micropipette and deposited in the appropriate well of the test cassette. COVID-PRESTO®'s performance has been evaluated in a recent clinical study, and the test showed a specificity of 100% and a sensitivity ranging from 10% (when the test was done 0–5 days after symptoms onset) to 100% (when performed ≥15 days after experiencing the first symptoms) [11]. Those results were obtained in a controlled setting where the test was performed by trained professionals. In order to be suitable as a self-test for the general population, COVID-PRESTO® has to be evaluated for practicability, i.e. its capacity of being correctly performed by untrained individuals. The aim of this study

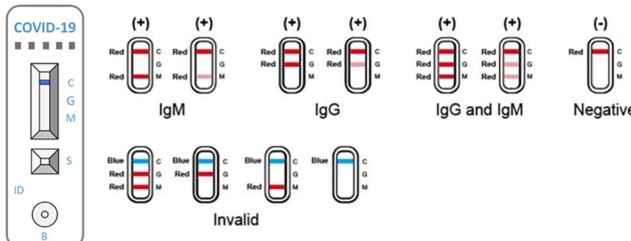

**Fig 1. Interpretation of results for COVID-PRESTO®.** COVID-PRESTO® uses anti-human immunoglobulin (Ig) M antibody (test line M), anti-human IgG antibody (test line G) and rabbit IgG (control line C) immobilized on a nitrocellulose strip. The Conjugate (recombinant COVID-19 antigens labeled with colloidal gold) is also integrated into the strip. A specimen is added to the sample well (S) followed by addition of assay buffer to the buffer well (B), IgM and/or IgG antibodies if present, will bind to COVID-19 conjugates forming antigen-antibody complexes. These complexes migrate through the nitrocellulose membrane by capillary action. When the complex meets the line of the corresponding immobilized antibody (anti-human IgM and/or anti-human IgG), the complex is trapped forming a burgundy colored band which confirms a reactive test result. The absence of a colored band in the test region indicates a non-reactive test result. To serve as a procedural control, a colored line always changes from blue to red in the control line region, indicating that the proper volume of specimen has been added and membrane wicking has occurred. If the control line does not switch colors, then the test is deemed invalid.

was to evaluate the COVID-PRESTO® test in terms of participants' capability to obtain a valid test, and the users' ability to interpret the results.

## Methods and materials

### Ethical approval

The study was approved by the Regional Orléans Research and Ethics Committee on March 17th 2020. Each participant was given an information leaflet and oral consent was obtained directly by the attending physician. Participants were given the choice to opt out of the study at any time. Participation in the study did not alter the treatment given to the subject (if any). All participants were treated according to the standard of care.

### Study population

Study volunteers were selected from four different locations in Central France from March 20th to May 5th, 2020: two nuclear plants (Dampierre-en-Burly and Saint Laurent-des-Eaux), individuals visiting the vaccination clinic of Orleans Regional Hospital and non-medical staff from Orleans Regional Hospital. The decision of recruiting non-medical staff exclusively was made in order to avoid bias due to previous or current experience in blood drawing. A total of 143 volunteers participated to the study. Two were excluded from the analysis because they did not fill in the questionnaire.

A 2-step design, similar to the design used to validate a HIV self-test, was used for this study [12].

**Substudy 1: Usage of the self-test.** Two different instructional media were available: written Instructions for Use and an instructional video. Each volunteer was invited to use either one or both, in order to get a clear understanding of how to perform the test.

The instructions were as follows (Table 1):

Every volunteer had to use the lancet needle to prick the side of the fingertip to let a large drop of suspended blood form, collect the drop with a 10 μl capillary micropipette that filled automatically, transfer the blood into the sample well and finally, add two drops of buffer in the buffer well.

Each participant was asked to fill in a satisfaction questionnaire regarding the suitability of COVID-PRESTO® as a future self-test intended for the general population (Table 2).

The test user was monitored by an observer (occupational health physician at the nuclear plants, trained nurse or physician at the Orleans Regional hospital) who could assist the user, if asked to, and evaluate the execution of the different tasks. This evaluation could be different from the user's personal feedback (Table 3).

**Substudy 2: Reading and interpretation of results.** Each participant was shown a basket containing 6 standardized test results (2 positives, 2 negatives and 2 invalids). The participant had to randomly choose three out of the six standardized tests and write down the results for each test. A supervisor was in charge of collecting the responses and assessing their accuracy.

### Data analysis

The population was described in terms of percentages. The influence of demographic characteristics such as age and education level, on the response to the questionnaire, users' satisfaction and ability to correctly perform and interpret the test were evaluated using univariate and stratified analyses and the Cochran-Mantel-Haenszel test. In case of overall differences between groups, *post hoc* Fisher or $\chi^2$ tests with Bonferroni correction were applied for two-group comparisons.

**Table 1. Instructions for use.**

| Autotest COVID® instructions for use |
| --- |
| **Summary and description of the test** |
| Autotest COVID® is a screening test for IgG and IgM antibodies against SARS-CoV-2 based on a blood sample taken from the tip of the finger. |
| Autotest COVID® allows for the determination of immunization against SARS-CoV-2 which makes it possible to assert, even in the absence of symptoms, contact with the virus and a supposed acquired protective immunity. |
| Autotest COVID® is a one-time use in-vitro diagnostic test. |
| Autotest COVID® is intended for use by a layperson in a private setting. |
| The test takes about 5 minutes to perform and the waiting time before reading the result is 10 minutes. |
| You will need a watch, clock, or other timing device. Please carefully read all of the following instructions prior to using the test. |
| **Content of the kit** |
| - Foil pouch containing the test device and a desiccant packet. |
| - One disinfectant wipe |
| - 1 vial of buffer |
| - 1 safety lancet |
| - 1 capillary blood pipette (10μl) |
| **Test procedure**: |
| Step 1: Clean the fingertip with the disinfectant wipe and let it dry |
| Step 2: Prick the fingertip using the safety lancet |
| Step 3: Press the fingertip to form a drop of blood |
| Step 4: Collect the blood sample using the capillary pipette |
| Step 5: Discharge the pipette into the Sample well "S" |
| Step 6: Put 2 drops of buffer into Buffer well "B" |
| Step 7: Read the result of the test after 10 minutes |

**Table 2. Satisfaction questionnaire.**

| Satisfaction Questionnaire |
| --- |
| **1**. Did you find the test's instructions for use provided to be clear and comprehensible? |
| ☐ Yes ☐ No |
| **2**. Did you find the instructional video to be clear and comprehensible? |
| ☐ Yes ☐ No |
| **3**. Did you manage to collect the recommended amount of blood? |
| ☐ Yes ☐ No |
| **4**. Did you manage to fill the pipette? |
| ☐ Yes ☐ No |
| **5**. Did you manage to deposit the blood and buffer in the wells without mistake? |
| ☐ Yes ☐ No |
| **6**. Did you obtain a valid result (at least one band)? |
| ☐ Yes ☐ No |
| **7**. What is your opinion regarding the use of this test as a self-test? |
| ☐ Bad ☐ Mediocre ☐ Good ☐ Very good ☐ Excellent |
| **8**. What is your opinion regarding the readability and the interpretation of this test? |
| ☐ Bad ☐ Mediocre ☐ Good ☐ Very good ☐ Excellent |

# Results

Overall, 143 volunteers were selected to participate in the study, among those, 2 did not fill in the Satisfaction questionnaire and 96 participated in Sub-study 2. The distribution of gender

**Table 3. Supervisor's feedback.**

| Supervisor's Feedback |
| --- |
| **1**. Was the supervisor's assistance requested by the participant? |
| ☐ Yes ☐ No |
| **2**. What is your opinion regarding the execution of this test by the participant? |
| ☐ Bad ☐ Mediocre ☐ Good ☐ Very good ☐ Excellent |

was biased towards men, who represented two thirds of the population. Four different age categories were defined: 20–29, 30–39, 40–49 and ≥50. Three education level categories were defined according to the International Classification of Education (2011 version): Level 3 or below, Level 4 or 5, and Level 6 or above.

The demographic characteristics are provided in Table 4.

## Substudy 1

A total of 104 participants read the instructions for use provided with the test. Among those, 70 volunteers read the instructions and watched the video. 92 participants (88.5%) found the written instructions to be clear and comprehensible. 107 participants watched the video and among those, 37 volunteers watched the video as the only instruction medium. 97/107 participants (90.7%) found the video to be clear and comprehensible (Table 5). A subgroup analysis showed that there was no difference on the instructions' comprehension for use when age or education level was considered. However, there was a statistical difference in the comprehension of the instructional video among the groups. Indeed the video was judged comprehensible by only 77.8% of the participants aged 29 or less compared to 91.7% (30–39), 96.7% (40–49) and 100.0% (≥50) by the other groups (Cochran-Mantel-Haenszel test's overall p = 0.0465).

**Table 4. Demographic characteristics of the study population.**

| Characteristics | Substudy 1 N = 141 | | Substudy 2 N = 96 | |
| --- | --- | --- | --- | --- |
| | N (%) | % | N (%) | % |
| **Study Site**: | | | | |
| Dampierre Nuclear Plant | 44 | 31.2 | 23 | 24.0 |
| St Laurent Nuclear Plant | 39 | 27.7 | 15 | 15.6 |
| Orleans Regional Hospital | 58 | 41.1 | 58 | 60.4 |
| **Age** | | | | |
| 20–29 | 32 | 22.7 | 25 | 26.0 |
| 30–39 | 45 | 31.9 | 30 | 31.3 |
| 40–49 | 45 | 31.9 | 29 | 30.2 |
| ≥ 50 | 19 | 13.5 | 12 | 12.5 |
| **Gender** | | | | |
| Male | 97 | 68.8 | 56 | 58.3 |
| Female | 44 | 31.2 | 40 | 41.7 |
| **Education Level**[*] | | | | |
| Level 3 or below | 24 | 17 | 18 | 18.8 |
| Level 4 or 5 | 75 | 53.2 | 48 | 50.0 |
| Level 6 or above | 42 | 29.8 | 30 | 31.3 |

[*]According to the International Standard Classification of Education 2011.

**Table 5. Instructions' comprehension.**

| | Written instructions comprehensible | | Video comprehensible | |
|---|---|---|---|---|
| | N = 104 | | N = 107 | |
| | **Yes** | **No** | **Yes** | **No** |
| **Total** | 92 | 12 | 97 | 10 |
| **Age** | | | | |
| 20–29 | 22 | 3 | 21 | 6 |
| 30–39 | 27 | 5 | 33 | 3 |
| 40–49 | 29 | 1 | 29 | 1 |
| 50+ | 14 | 3 | 14 | 0 |
| Cochran-Mantel-Haenszel test | 0.3766 | | **0.0465** | |
| **Education Level** | | | | |
| 3 or below | 21 | 1 | 16 | 4 |
| 4/5 | 40 | 8 | 53 | 1 |
| 6 or above | 31 | 3 | 28 | 5 |
| Cochran-Mantel-Haenszel test | 0.2848 | | **0.0235** | |

However, *post hoc* Fisher's tests with Bonferroni correction revealed that the proportion of users that found the video comprehensible in the 20–29 range was not significantly lower when groups were compared by pairs, meaning that age has little influence on the comprehension of the instructional video. There was also an overall statistical difference in the comprehension of the video when education level was considered as an influencing factor (Cochran-Mantel-Haenszel test's overall p = 0.0235). Nevertheless, when two-group comparisons were performed using a Fisher's t test or a $\chi^2$ with a Bonferroni correction, no difference was observed. These results show that education level did not have an effect on the understanding of the instructional video. Taken together these results show that the instructions were well received and understood by all participants with little to no impact of age or education level on their comprehension.

The COVID-PRESTO® technical practicability was assessed by analyzing the participants' responses to the satisfaction questionnaire. 130 out of the 141 participants (92.2%) who filled in the questionnaire, were able to draw enough blood using the lancet needle on their fingertips. However, the ten participants that could not collect the recommended quantity of blood were able to proceed with the subsequent steps. 128 participants (90.8%) succeeded in filling up the pipette with blood on the first try and without assistance. 135 testers (95.7%) were able to correctly deposit both the blood and the buffer in their respective wells. Among those, 7 participants needed assistance to fill the pipette. At the end of the procedure, nearly all participants (139/141, 98.6%) declared that they had been able to get a valid result (defined by a pink band in the control lane).

In terms of overall satisfaction, 133/141 participants (94.3%) rated the COVID-PRESTO® positively (defined as "Good", "Very Good" or "Excellent"), based on the ease of execution of the test's procedures. 45.4% of the participants rated the test as good and 49.6% as "Very Good" or "Excellent" (Fig 2). A subgroup analysis revealed that age and education had no impact whatsoever on the overall satisfaction with the COVID-PRESTO® test.

Supervisors were also asked to answer a survey after each participant was done with the test's procedures. It is, however, important to point out that only 122 answers to the supervision survey were collected. During these supervised sessions, 92/122 (76.4%) participants did not ask for the supervisor's assistance. Only 30 participants requested assistance mainly for

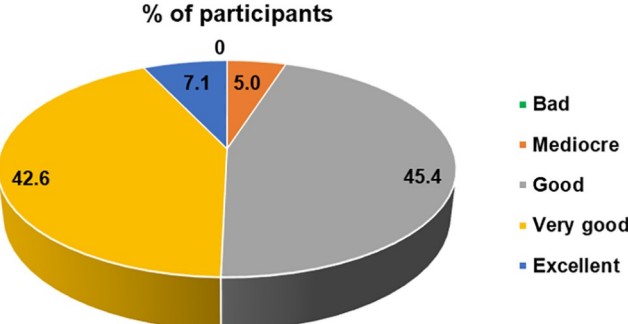

**Fig 2. User satisfaction regarding the use of COVID-PRESTO® as a self-test.** Percentages of participants for each choice of response to the question no.7 of the satisfaction questionnaire.

tips on how to take off the cap of the needle as well as on how to use the pipette. The subgroup analysis showed that almost half of the participants (43.8%) with an education level of 4 or 5 requested assistance and this proportion was statistically higher when compared to the proportion of participants having requested assistance in the other groups (Fig 3, overall Cochran-Mantel-Haenszel test's p = 0.0004; $\chi^2$ p value of 0.009 and 0.006 vs. level ≤ 3 and ≥ 6 respectively). The supervisors' opinion on the execution of the procedures by the participants was collected and the execution was rated as good or better for 113/122 (92.7%) volunteers. 45.9% of the volunteers were rated as "Good", 36.1% as "Very Good" and 10.7% as "Excellent". When age and education parameters were considered during a subgroup analysis, no difference was observed between the categories in terms of execution ratings (Fig 4A and 4B).

## Substudy 2

96 volunteers out of the 141 participants in the first phase took part in this second phase. The volunteers had to randomly choose three tests and write down the results. A supervisor was in charge of collecting and assessing their responses. Among these 96 participants, 94 (97.9%) judged the readability of the test to be good or better ("very good" or "excellent"). 41.2% of the volunteers rated the test legibility as "Good", 41.7% as "Very Good" and 14.5% as "Excellent". Only two individuals incorrectly interpreted the test by judging an invalid test as valid. The

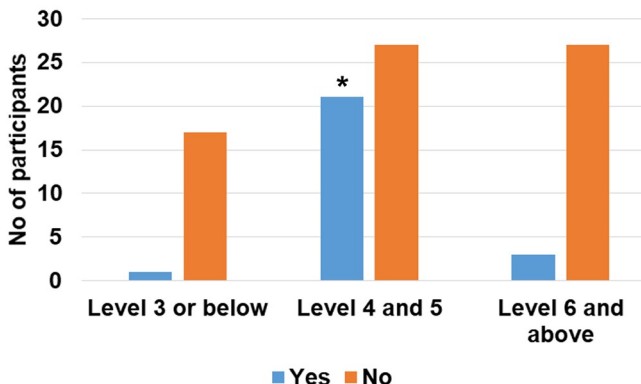

**Fig 3. Participants with an education level 4 or 5 requested assistance more frequently.** Number of participants having asked for a supervisor's assistance according to education level (* $\chi^2$ p<0.05 vs. education level 3 and below or level 6 and above).

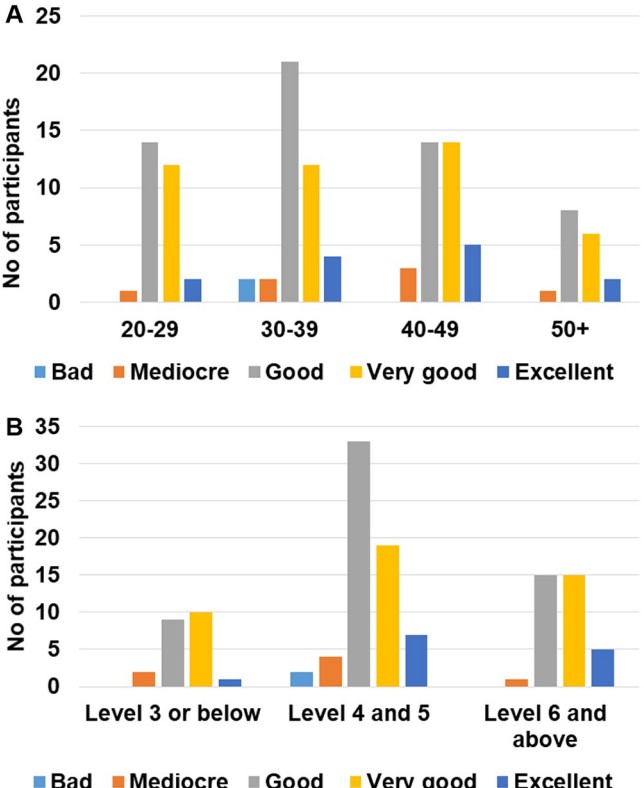

**Fig 4. Description of the execution ratings according to the supervisors.** Number of participants in each rating category according to age (A) or education level (B).

confusion originated from the fact that they had not understood that the control lane had to be pink instead of blue. Overall, 288 tests were read and only 2 were not interpreted correctly, resulting in a 99.3% success rate. These two participants were 26 and 42 years old and with an educational level of 3 and 4 respectively. A subgroup analysis considering the age and education level showed that these two parameters don't have any influence on the ability of the participant to correctly interpret the test results (Fig 5A and 5B).

## Discussion

The objective of this study was to test the adequacy of one Rapid Diagnosis Test for Covid-19 in regard to a potential release for the general population. This test was developed to detect the presence of antibodies targeted against the SARS-CoV-2 as an indirect marker of prior infection.

The use of such tests, with the help of appropriate instructions and interpretation guidelines, could prove to be essential in supporting the current public health effort. Indeed, these tests would provide data on the dissemination of virus across the population and would therefore be a valuable tool to help comprehend the epidemic. The use of such tests will also be essential to know the immune status of patients in view of a second wave or a possible mass vaccination program [10].

COVID-PRESTO® has recently been evaluated for performance and showed sensitivity ranging from 69% for patients with symptoms that occurred from 11 to 15 days before the date of test to 100% in patients who experienced first symptoms more than 15 days before the

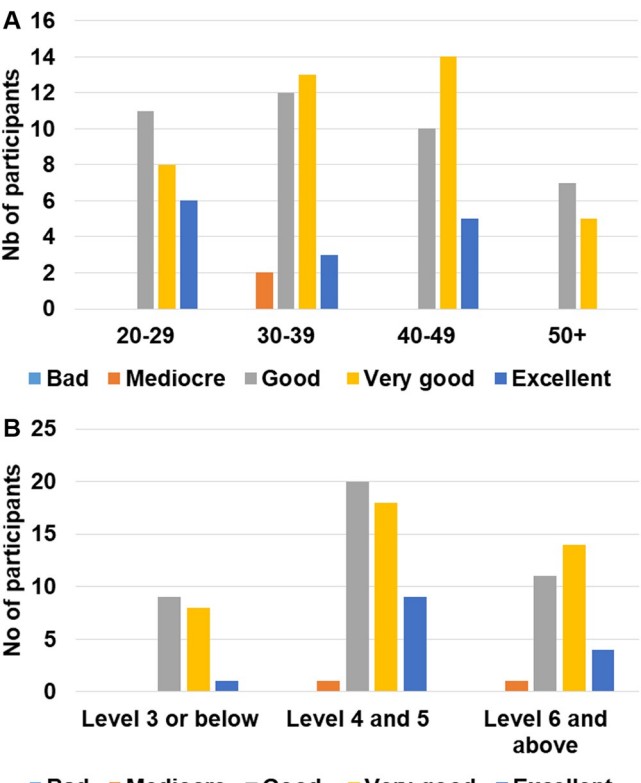

**Fig 5. Description of the readability of the test results according to the participants.** Number of participants in each rating category according to age (A) or education level (B).

test [13]. However, for the test to be approved as a self-test, a practicability study was needed in order to assess the feasibility of the test for use by untrained individuals with different education levels. This study showed that the instruction materials provided with the test are clear and comprehensible regardless of the user's age or education level. However, our results showed that age or education level may slightly influence the comprehension of non-written instructions, i.e. instructional video. This finding indicates that it may be necessary to include a variety of instruction media along with the test to ensure its comprehension by a broader audience. These media may include written instructions, cartoons or videos.

The practicability of an assay is determined by the ability of the end user to obtain a valid result and interpret it. Regarding both of these criteria, the COVID-PRESTO® performed well. Indeed our results showed that 92% of the participants performed the test correctly as assessed by trained supervisors, and randomly-chosen tests were interpreted correctly by 99% of the participants. The study revealed that neither age nor the education level had an influence on either the ability to correctly read the test or the ability to execute the procedures. This strongly suggests that the execution of the test is accessible to a wide range of persons. Nevertheless, we observed that the education level influenced whether or not the participant asked for assistance. Finally, this study revealed that the COVID-PRESTO® is judged practical with a global satisfaction rate of 95% and is favorably seen as a potential self-test by the users. Taken together, these results are in favor of the COVID-PRESTO®'s potential to be considered as a self–test.

Data are scarce regarding the performance of the existing serological COVID-19 tests, and to our knowledge this is the first practicability study regarding a COVID-19 POC test, making

it difficult to compare our results with other devices. However, this kind of study is necessary before making a self-test available to the general public in order to, on the one hand, avoid confusion about false positive results thus leading to unnecessary demands on health services and, on the other hand, avoid false negative results potentially leading to an underestimation of the virus' presence across the population.

The ease of understanding the instructions is always a challenge when designing a self-test, but our study shows that the COVID-PRESTO® test procedures fare well in that regard. However, feedback from users showed that there is still room for improvement regarding the instructions and video. Indeed, some of the volunteers had legitimate questions on technical procedures such as the handling of the lancet needle, and the use of the pipette. This was shown by the fact that almost a quarter of the participants asked for assistance from a supervisor. It is important to point out that in our study, socio-demographic parameters, such as age and education, did not influence any of the tested parameters. This suggests that the instructions as well as the procedures are sufficiently clear and simple to be executed by the general public.

The next step for a person that has been tested positive for the Covid-19 is knowing if he/she is still contagious and whether he/she has to isolate his/herself from others. It has been recently demonstrated that individuals who have recovered form a mild form of the COVID-19 possess neutralizing antibodies [10], however, based on the currently available data we can only assume without certainty that infection with the virus might generate protective immunity [14]. It is therefore important that the test instructions include clear and understandable guidance regarding the actions that need to be taken in case of a positive or negative result, in order to avoid a relaxation of safety measures. To this end, the instructions for use of the COVID-PRESTO® test will recommend seeking care from a Primary Care provider and undergoing further testing (PCR) to confirm/invalidate the presence of an active infection in case of a positive test.

## Conclusion

These 2 substudies indicate that the finger-stick COVID PRESTO® self-test is practical and that test users correctly read the results. The COVID-PRESTO® should be considered as a suitable candidate for a public release in order to provide an additional tool to gather information about the dissemination of the virus across the population.

## Supporting information

**S1 Dataset.**
(XLSX)

## Acknowledgments

The authors would like to thank the technical staff of the Department of Infectious diseases for their excellent assistance. Furthermore, the authors thank Thibaut de Sablet of Clinact, France for providing medical writing support/editorial support in accordance with Good Publication Practice (GPP3) guidelines.

## Author Contributions

**Conceptualization:** Thierry Prazuck, Gilles Pialoux.

**Formal analysis:** Thierry Prazuck, Gilles Pialoux.

**Funding acquisition:** Thierry Prazuck.

**Investigation:** Thierry Prazuck, Jean Phan Van, Florence Sinturel, Frederique Levray, Allan Elie, Denise Camera.

**Methodology:** Thierry Prazuck, Gilles Pialoux.

**Project administration:** Thierry Prazuck.

**Supervision:** Thierry Prazuck, Jean Phan Van, Florence Sinturel, Frederique Levray, Allan Elie, Denise Camera.

**Validation:** Thierry Prazuck, Gilles Pialoux.

**Visualization:** Thierry Prazuck, Gilles Pialoux.

**Writing – original draft:** Thierry Prazuck, Gilles Pialoux.

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
