## [Decision Letter · Decision Letter 0]

24 Sep 2020

PONE-D-20-21033

Evaluation of the Practicability of a Finger-Stick Whole-Blood SARS-Cov-2 Self-Test Adapted for the General Population.

PLOS ONE

Dear Dr. Prazuck,

Thank you for submitting your manuscript to PLOS ONE. After careful consideration, we feel that it has merit but does not fully meet PLOS ONE’s publication criteria as it currently stands. Therefore, we invite you to submit a revised version of the manuscript that addresses the points raised during the review process.

Your manuscript was seen by two experts in the field. Both reviewers have raised several issues and concerns that should be fully addressed in a revised manuscript. 

We look forward to receiving your revised manuscript.

Kind regards,

Sanjai Kumar

Academic Editor

PLOS ONE

Journal Requirements:

2. Please provide additional details regarding participant consent. In the ethics statement in the Methods and online submission information, please ensure that you have specified (1) whether consent was informed and (2) how oral consent was documented and witnessed. If your study included minors, state whether you obtained consent from parents or guardians.

3. In your methods section, please provide the names of the four different locations volunteers were selected from.

4. Please discuss whether participants were able to opt out of the study and whether individuals who did not participate received the same treatment offered to participants.

5.We note that you have indicated that data from this study are available upon request. PLOS only allows data to be available upon request if there are legal or ethical restrictions on sharing data publicly. For information on unacceptable data access restrictions, please see http://journals.plos.org/plosone/s/data-availability#loc-unacceptable-data-access-restrictions.

6. We note you have included a table to which you do not refer in the text of your manuscript. Please ensure that you refer to Table 1 in your text; if accepted, production will need this reference to link the reader to the Table.

Reviewers' comments:

Reviewer's Responses to Questions

**Comments to the Author**

1. Is the manuscript technically sound, and do the data support the conclusions?

Reviewer #1: Partly

Reviewer #2: Yes

2. Has the statistical analysis been performed appropriately and rigorously? 

Reviewer #1: Yes

Reviewer #2: I Don't Know

3. Have the authors made all data underlying the findings in their manuscript fully available?

Reviewer #1: Yes

Reviewer #2: Yes

4. Is the manuscript presented in an intelligible fashion and written in standard English?

Reviewer #1: No

Reviewer #2: Yes

5. Review Comments to the Author

Reviewer #1: The manuscript “Evaluation of the Practicability of a Finger-Stick Whole-Blood SARS-CoV-2 Self-Test Adapted for the General Population” describes the results of a usability study conducted to evaluate the suitability of use of the COVID-PRESTO rapid test by the general population. The COVID-PRESTO test is a lateral flow chromatographic immunoassay that has been developed to detect and differentiate IgG and IgM antibodies to SARS-CoV-2 in human fingerstick whole blood. The study aims to assess the practicability of the test when used in non-laboratory settings. The study included 142 participants with no experience in blood drawing, whose capability to obtain an interpretable result and ability to read the results were evaluated. 88.4 % of the users judged the instructions for use to be clear and understandable. 99.3 % of the users obtained a valid result and, according to the supervisors, 92.7% of the tests were properly performed by the user. The results suggest that the test shows a potential to be used as a self-test.

Major comments:

1. The authors refer to the test used in the study as a “rapid diagnostic test”. However, generally, a diagnostic test is a test that detects an acute infection, i.e. an RT-PCR or an antigen test. Antibody assays identify individuals with an adaptive immune response, which indicates prior rather than acute infection.

2. Lines 230 – 233: it is stated that “Around a hundred of COVID-19 serological tests are currently commercially available in the US through the Emergency Use Authorization (EUA) granted to the US Centers for Disease Control and Prevention by the Food and Drug Administration (FDA)”. This statement is not correct. In fact, an EUA is granted by FDA to every manufacturer of a test, not to CDC, and as of 08/09/2020, there are 35 FDA-authorized serological tests (https://www.fda.gov/medical-devices/coronavirus-disease-2019-covid-19-emergency-use-authorizations-medical-devices/vitro-diagnostics-euas#individual-serological).

3. In Discussion (lines 275 – 276), a conclusion is made that the COVID-PRESTO is judged practical with a global satisfaction rate of 95% by the users and is favorably seen as a potential self-test. I believe that users’ satisfaction rate is not an indicator of a test’s suitability to be used outside professional healthcare facilities. The practicability of an assay as a self-test should be determined by the ability of the end user to correctly perform the test, as evaluated by a trained professional. A satisfaction questionnaire can be used to evaluate the ease of understanding the instructions for use.

4. The main conclusion that the authors make is that the COVID-PRESTO assay “should be considered as a suitable candidate for a public release”. I think it should also be noted that improvements need to be made to the test’s instructions for use (or other mitigating measures need to be put in place) because 24.6% of participants asked for assistance when administering the test.

5. In addition, although the study results seem to suggest that the test may be suitable as a self-test for the general population, the authors have to keep in mind that there may be different regulatory requirements in different parts of the world that need to be met in order for the test to be approved for this specific use.

Minor comments:

Consulting a professional editor will improve the manuscript’s readability by minimizing grammatical inaccuracies.

Reviewer #2: SUMMARY

In this study, the authors evaluate the practicability of the AAZ COVID-PRESTO® test, a Rapid Diagnostic Test (RDT), easy-to-use device to detect specific antibodies to SARS-CoV-2 in human whole blood.

In late 2019 the emergence of SARS-CoV-2, which causes COVID-19, initiated a pandemic that has been associated with over 500,000 deaths globally as of July 2020. The original outbreak that started in Wuhan, China, rapidly expanded into other continents.

AAZ COVID-PRESTO® is a lateral flow immunochromatographic assay that can be conducted using a single drop of blood. The test uses anti-human IgM antibody (test line IgM), anti-human IgG antibody (test line IgG) and rabbit IgG (control line C) immobilized on a nitrocellulose strip. The Conjugate (recombinant COVID-19 antigens labeled with colloidal gold) is also integrated into the strip.

When a specimen is added to the sample well (S) followed by assay buffer added to the buffer well (B), IgM &/or IgG antibodies if present, will bind to COVID-19 conjugates forming immunocomplexes. These complexes migrate through the nitrocellulose membrane by capillary action. When complexes meet the line of the corresponding immobilized antibody (anti-human IgM &/or anti-human IgG), they are captured forming a burgundy colored band which confirm a reactive test result. Absence of a colored band in the test region indicates a non-reactive test result. If the proper volume of specimen has been added, a colored line will change from blue to red in the control line region.

The authors describe a study conducted in 142 volunteers selected from four different locations in Central France from March 20th through May 5th, 2020. Each volunteer had to perform the following tasks:(i) read test’s instructions or watch instructional video, (ii) use the lancet to prick the fingertip and extract a drop of blood, (iii) conduct the test, (iv) read and interpret test results and (v) fill out a questionnaire.

Finally, the authors concluded that the COVID-PRESTO® assay was successfully used by most participants and showed a promising potential as a self-test to be used by the general population.

MERITS

In late 2019 the emergence of SARS-CoV-2, which causes COVID-19, initiated a pandemic that has been associated with over 500,000 deaths globally as of July 2020. The original outbreak that started in Wuhan, China, rapidly expanded into other continents. In United States, novel coronavirus has spread to nearly every state and territory, accounting for at least 192,000 deaths since February 2020.

The topic of this manuscript is extremely relevant in the current stage of the pandemic.

Containment strategies have focused on travel restrictions, isolation, use of masks, and contact tracing. As governments start easing these rules, and citizens return to daily but restricted activities, health and research authorities are searching for effective ways to test, control and track every case in large populations.

In this context, the development of new easy-to-use tests that could be accessible and distributed to everyone is imperative to help control the spread of SARS-CoV-2. After a week from the first clinical manifestations, the sensitivity of RT-PCR tests diminishes gradually, due to the decreasing amount of virus particles in the respiratory tract epithelium. In such cases, patients may have false negative results, despite the ongoing infection. IgG and IgM antibodies against the virus can be detected with 1-3 weeks after exposure.

Serological self-administered tests are of great interest, to assess the degree of immunization, to detect asymptomatic cases, to trace contacts, and to support decisions to re-admit people at work, schools and universities. Indeed, these tests could become crucial in supporting public health efforts to track virus spread and to develop containment strategies.

This work has several strengths:

• It provides additional studies of the suitability of the COVID-PRESTO® test needed to show its performance in untrained individuals as a self-administered assay. These studies are required by the European Centre for Disease Prevention and Control, before a test can be made available to the public.

• The assay was tested in non-medical volunteers, to avoid bias due to (i) previous or

current experience in blood drawing and (ii) familiarity with interpretation of diagnostic tests.

• The test’s instructions can be followed in two different formats (printed or video), which expands accessibility to materials.

• The results showed that the execution of the test was easily conducted by a wide range of individuals independently of age or education level without the need of previous training.

• This rapid test has the potential to be used for screening in low resource or emergency settings, such as evacuation or refugee centers.

CRITIQUE

1. Introduction

a. This section provides a generalized background of the topic. The authors may wish to provide a brief description of the AAZ COVID-PRESTO®, how it’s executed and indicate some of its characteristics, including specificity and sensitivity.

b. The paper that you reference on Page 3, line 92, has been published. The sentence starting “It has been evaluated …” (Line 90) will need to be edited to reflect that.

c. Interpretation of Figure 1 will benefit of a caption with a brief explanation. In addition, please define all abbreviations used.

d. Minor edits

Please correct the following sentences:

(d1) Line 75: add a coma between resource center and the COVID-19 pandemics.

(d2) Line 77: add a stop after health emergency and start the next sentence as “The mark of…”

(d3) Line 81: change the sentence from “In most countries these measures have been effective in slowing down the transmission of the virus” to “In most countries these measures have been effective in slowing down virus transmission”.

(d4) Line 96: change the sentence from “The aim of this study was to evaluate the COVID-PRESTO® test in terms of participants’ capability to obtain an interpretable result, and the users’ ability to interpret the results” to “The aim of this study was to evaluate the COVID-PRESTO® assay in terms of participants’ capability to obtain a valid test, and the users’ ability to interpret the results”.

2. Methods

e. The first sentence of the Data Analysis section needs clarification. It’s not possible to understand how the authors described the statistical analysis of the populations.

f. Minor edits

Please correct the following:

(f1) Table 1 needs formatting. Please move the title to the empty row and add a section header (Summary and explanation of the test) so that it aligns with the format you used in subsequent sections of your table.

(f2) Section 1, Table 1: capitalize the first word of each sentence.

(f3) Sub-study 2 section: the last two sentences need editing. Please consider changing them to “The participant had to randomly choose three tests and write down the results. A supervisor was in charge collecting and assessing their responses”.

(f4) Table 2, question 3: change to “Did you manage to collect the recommended amount of blood?”

(f5) Table 2, question 6: change to “Did you obtain a valid result (at least one band)?”

(f6) Table 3: please change the title to “Supervisor’s Feedback”.

3. Results

The results overall support the finding that the COVID-PRESTO® assay is an easy-to-use test and has promising potential to be used by the general population.

g. Table 4

(g1) Please explain what’s the difference between N(%) and % (under characteristics) and add a legend to the table.

(g2) In Sub-study 1, you indicate that N=142. However, only in the age category you have 142 volunteers while under study site, age and education level you have a sample of 140. Please explain the reason for this discrepancy and make changes to the totals if needed.

(g3) Sub-studies 1 and 2: Although the results are explained in the next, it is difficult to follow the outcomes related to the different groups and tasks you tested. Bar graphs, pie charts or additional tables could be helpful to better illustrate the findings and help the reader follow the analysis.

h. Minor edits

Please correct the following:

(h1) Line 147: change “to Sub-study 2” to “in Sub-study 2”.

(h2) Lines 148 and 149: change “classes” to categories”.

(h3) Line 149: the sentence should read “... level categories were defined according to the International Classification of Education …”.

(h4) Line 195: change “assistance to the supervisor” to “supervisor’s assistance”.

(h5) Line 216: change “…any influence of the ability of the participant…” to “…any influence in the ability of the participant…

4. Discussion

Findings are discussed in the context of the published literature and the references cited are relevant to the study. The data supports the conclusions of the work.

The discussion is long. The paragraphs related to the impact that the serology tests can have in fighting the current epidemic, regulations imposed by FDA, CDC and European health authorities and the need to assess the practicability of self-administered assays before promoting their use, belongs to the introduction of the paper.

g. Minor edits

(g1) Line 262: delete “These results are currently…” and add the reference after the previous sentence.

6. PLOS authors have the option to publish the peer review history of their article (what does this mean?). If published, this will include your full peer review and any attached files.

Reviewer #1: No

Reviewer #2: No

---

## [Author Response · Author response to Decision Letter 0]

1 Dec 2020

PONE-D-20-21033

Evaluation of the Practicability of a Finger-Stick Whole-Blood SARS-Cov-2 Self-Test Adapted for the General Population.

Dear PLoS one editors and reviewers,

The authors would like to thank you all for your time and expertise you brought into this review. Please find below our responses (in blue) to every of your remarks and editing requests. 

We hope that, following this revision, you will find that this manuscript reaches the scientific standards of publication in PLoSOne.

PLOS ONE

We have amended the manuscript following this editorial guidelines

2. Please provide additional details regarding participant consent. In the ethics statement in the Methods and online submission information, please ensure that you have specified (1) whether consent was informed and (2) how oral consent was documented and witnessed. If your study included minors, state whether you obtained consent from parents or guardians.

A sentence was added (line 150) to specify how consent was obtained. This study did not include minors.

3. In your methods section, please provide the names of the four different locations volunteers were selected from.

The names of the nuclear plants were added (line 157), the other two (Vaccination clinic of the CHR Orleans and The CHR Orleans) were already mentioned.

4. Please discuss whether participants were able to opt out of the study and whether individuals who did not participate received the same treatment offered to participants.

Most of the patients were healthy volunteers as this study is not about treatment. However, when applicable, participants were treated according to the standard of care.

5.We note that you have indicated that data from this study are available upon request. PLOS only allows data to be available upon request if there are legal or ethical restrictions on sharing data publicly. For information on unacceptable data access restrictions, please see http://journals.plos.org/plosone/s/data-availability#loc-unacceptable-data-access-restrictions.

The Database is made available as a supplementary information file

6. We note you have included a table to which you do not refer in the text of your manuscript. Please ensure that you refer to Table 1 in your text; if accepted, production will need this reference to link the reader to the Table.

Reference to Table 1 was added line 168

Reviewers' comments:

Comments to the Author

Reviewer #1:

 The manuscript “Evaluation of the Practicability of a Finger-Stick Whole-Blood SARS-CoV-2 Self-Test Adapted for the General Population” describes the results of a usability study conducted to evaluate the suitability of use of the COVID-PRESTO rapid test by the general population. The COVID-PRESTO test is a lateral flow chromatographic immunoassay that has been developed to detect and differentiate IgG and IgM antibodies to SARS-CoV-2 in human fingerstick whole blood. The study aims to assess the practicability of the test when used in non-laboratory settings. The study included 142 participants with no experience in blood drawing, whose capability to obtain an interpretable result and ability to read the results were evaluated. 88.4 % of the users judged the instructions for use to be clear and understandable. 99.3 % of the users obtained a valid result and, according to the supervisors, 92.7% of the tests were properly performed by the user. The results suggest that the test shows a potential to be used as a self-test.

Major comments:1. The authors refer to the test used in the study as a “rapid diagnostic test”. However, generally, a diagnostic test is a test that detects an acute infection, i.e. an RT-PCR or an antigen test. Antibody assays identify individuals with an adaptive immune response, which indicates prior rather than acute infection.

Agreed, we changed the nature of test to “Point of Care test, POC” throughout the document. This is consistent with the published performance paper as well.

2. Lines 230 – 233: it is stated that “Around a hundred of COVID-19 serological tests are currently commercially available in the US through the Emergency Use Authorization (EUA) granted to the US Centers for Disease Control and Prevention by the Food and Drug Administration (FDA)”. This statement is not correct. In fact, an EUA is granted by FDA to every manufacturer of a test, not to CDC, and as of 08/09/2020, there are 35 FDA-authorized serological tests (https://www.fda.gov/medical-devices/coronavirus-disease-2019-covid-19-emergency-use-authorizations-medical-devices/vitro-diagnostics-euas#individual-serological).

This reference has been cross checked and we have modified this statement in the text to reflect the reviewer’s comment in (lines 83 – 100)

3. In Discussion (lines 275 – 276), a conclusion is made that the COVID-PRESTO is judged practical with a global satisfaction rate of 95% by the users and is favorably seen as a potential self-test. I believe that users’ satisfaction rate is not an indicator of a test’s suitability to be used outside professional healthcare facilities. The practicability of an assay as a self-test should be determined by the ability of the end user to correctly perform the test, as evaluated by a trained professional. A satisfaction questionnaire can be used to evaluate the ease of understanding the instructions for use.

This point has been added in the discussion (lines 343 – 350). The ability of the user to correctly interpret the results has been tested in our study (sub-study 2) and the outcome was clearly positive (99% of correct answers).

4. The main conclusion that the authors make is that the COVID-PRESTO assay “should be considered as a suitable candidate for a public release”. I think it should also be noted that improvements need to be made to the test’s instructions for use (or other mitigating measures need to be put in place) because 24.6% of participants asked for assistance when administering the test.

This point is discussed in the paper (lines 366 – 375), and we acknowledge that changes need to be made in the instruction materials to increase their comprehension. The instruction leaflet and video will be amended to address the points that were considered not clear by the users. 

5. In addition, although the study results seem to suggest that the test may be suitable as a self-test for the general population, the authors have to keep in mind that there may be different regulatory requirements in different parts of the world that need to be met in order for the test to be approved for this specific use.

The conclusion of the paper is that COVID-PRESTO should be considered as a candidate for self-testing. Currently, this test is intended for the French market only. The French competent authority considers that it is premature to promote the use of SARs CoV-2 self-test due to a lack or published data regarding the reliability of these tests in real life conditions. Cf below (document in French):

https://www.has-sante.fr/upload/docs/application/pdf/2020-05/rapport_tests_serologiques_rapides_covid-19_vd.pdf

We hope that the results presented here will help fill that void and fuel the discussion about these tests.

Minor comments: Consulting a professional editor will improve the manuscript’s readability by minimizing grammatical inaccuracies.

The manuscript has been reviewed by a native English speaker in order to correct the grammatical inaccuracies

Reviewer #2: SUMMARY

In this study, the authors evaluate the practicability of the AAZ COVID-PRESTO® test, a Rapid Diagnostic Test (RDT), easy-to-use device to detect specific antibodies to SARS-CoV-2 in human whole blood.

In late 2019 the emergence of SARS-CoV-2, which causes COVID-19, initiated a pandemic that has been associated with over 500,000 deaths globally as of July 2020. The original outbreak that started in Wuhan, China, rapidly expanded into other continents.

AAZ COVID-PRESTO® is a lateral flow immunochromatographic assay that can be conducted using a single drop of blood. The test uses anti-human IgM antibody (test line IgM), anti-human IgG antibody (test line IgG) and rabbit IgG (control line C) immobilized on a nitrocellulose strip. The Conjugate (recombinant COVID-19 antigens labeled with colloidal gold) is also integrated into the strip.

When a specimen is added to the sample well (S) followed by assay buffer added to the buffer well (B), IgM &/or IgG antibodies if present, will bind to COVID-19 conjugates forming immunocomplexes. These complexes migrate through the nitrocellulose membrane by capillary action. When complexes meet the line of the corresponding immobilized antibody (anti-human IgM &/or anti-human IgG), they are captured forming a burgundy colored band which confirm a reactive test result. Absence of a colored band in the test region indicates a non-reactive test result. If the proper volume of specimen has been added, a colored line will change from blue to red in the control line region.

The authors describe a study conducted in 142 volunteers selected from four different locations in Central France from March 20th through May 5th, 2020. Each volunteer had to perform the following tasks:(i) read test’s instructions or watch instructional video, (ii) use the lancet to prick the fingertip and extract a drop of blood, (iii) conduct the test, (iv) read and interpret test results and (v) fill out a questionnaire.

Finally, the authors concluded that the COVID-PRESTO® assay was successfully used by most participants and showed a promising potential as a self-test to be used by the general population.

MERITS

In late 2019 the emergence of SARS-CoV-2, which causes COVID-19, initiated a pandemic that has been associated with over 500,000 deaths globally as of July 2020. The original outbreak that started in Wuhan, China, rapidly expanded into other continents. In United States, novel coronavirus has spread to nearly every state and territory, accounting for at least 192,000 deaths since February 2020.

The topic of this manuscript is extremely relevant in the current stage of the pandemic.

Containment strategies have focused on travel restrictions, isolation, use of masks, and contact tracing. As governments start easing these rules, and citizens return to daily but restricted activities, health and research authorities are searching for effective ways to test, control and track every case in large populations.

In this context, the development of new easy-to-use tests that could be accessible and distributed to everyone is imperative to help control the spread of SARS-CoV-2. After a week from the first clinical manifestations, the sensitivity of RT-PCR tests diminishes gradually, due to the decreasing amount of virus particles in the respiratory tract epithelium. In such cases, patients may have false negative results, despite the ongoing infection. IgG and IgM antibodies against the virus can be detected with 1-3 weeks after exposure.

Serological self-administered tests are of great interest, to assess the degree of immunization, to detect asymptomatic cases, to trace contacts, and to support decisions to re-admit people at work, schools and universities. Indeed, these tests could become crucial in supporting public health efforts to track virus spread and to develop containment strategies.

This work has several strengths:

• It provides additional studies of the suitability of the COVID-PRESTO® test needed to show its performance in untrained individuals as a self-administered assay. These studies are required by the European Centre for Disease Prevention and Control, before a test can be made available to the public.

• The assay was tested in non-medical volunteers, to avoid bias due to (i) previous or current experience in blood drawing and (ii) familiarity with interpretation of diagnostic tests.

• The test’s instructions can be followed in two different formats (printed or video), which expands accessibility to materials.

• The results showed that the execution of the test was easily conducted by a wide range of individuals independently of age or education level without the need of previous training.

• This rapid test has the potential to be used for screening in low resource or emergency settings, such as evacuation or refugee centers.

CRITIQUE

1. Introduction

a. This section provides a generalized background of the topic. The authors may wish to provide a brief description of the AAZ COVID-PRESTO®, how it’s executed and indicate some of its characteristics, including specificity and sensitivity.

Added: lines 114 to 121

b. The paper that you reference on Page 3, line 92, has been published. The sentence starting “It has been evaluated …” (Line 90) will need to be edited to reflect that.

Corrected: lines 118 to 121

c. Interpretation of Figure 1 will benefit of a caption with a brief explanation. In addition, please define all abbreviations used.

An explanation has been added in lines 130 to 143

d. Minor edits

Please correct the following sentences:

(d1) Line 75: add a coma between resource center and the COVID-19 pandemics.

Corrected

(d2) Line 77: add a stop after health emergency and start the next sentence as “The mark of…”

Corrected

(d3) Line 81: change the sentence from “In most countries these measures have been effective in slowing down the transmission of the virus” to “In most countries these measures have been effective in slowing down virus transmission”.

Corrected

(d4) Line 96: change the sentence from “The aim of this study was to evaluate the COVID-PRESTO® test in terms of participants’ capability to obtain an interpretable result, and the users’ ability to interpret the results” to “The aim of this study was to evaluate the COVID-PRESTO® assay in terms of participants’ capability to obtain a valid test, and the users’ ability to interpret the results”.

Done

2. Methods

e. The first sentence of the Data Analysis section needs clarification. It’s not possible to understand how the authors described the statistical analysis of the populations.

We rephrased this section in order to clarify our methods. 

f. Minor edits

Please correct the following:

(f1) Table 1 needs formatting. Please move the title to the empty row and add a section header (Summary and explanation of the test) so that it aligns with the format you used in subsequent sections of your table.

Corrected

(f2) Section 1, Table 1: capitalize the first word of each sentence.

Corrected

(f3) Sub-study 2 section: the last two sentences need editing. Please consider changing them to “The participant had to randomly choose three tests and write down the results. A supervisor was in charge collecting and assessing their responses”.

Corrected

 (f4) Table 2, question 3: change to “Did you manage to collect the recommended amount of blood?”

Corrected

(f5) Table 2, question 6: change to “Did you obtain a valid result (at least one band)?”

Corrected

(f6) Table 3: please change the title to “Supervisor’s Feedback”.

Corrected

3. Results

The results overall support the finding that the COVID-PRESTO® assay is an easy-to-use test and has promising potential to be used by the general population.

g. Table 4

(g1) Please explain what’s the difference between N(%) and % (under characteristics) and add a legend to the table.

It was a mistake, there was no difference, the table has been corrected accordingly.

(g2) In Sub-study 1, you indicate that N=142. However, only in the age category you have 142 volunteers while under study site, age and education level you have a sample of 140. Please explain the reason for this discrepancy and make changes to the totals if needed.

The entire table has been corrected. Errors were spotted after initial submission of the manuscript to PLoSOne. We addressed them in the revised manuscript. Proportions changed a little but nothing significant was observed.

(g3) Sub-studies 1 and 2: Although the results are explained in the next, it is difficult to follow the outcomes related to the different groups and tasks you tested. Bar graphs, pie charts or additional tables could be helpful to better illustrate the findings and help the reader follow the analysis.

Bar graphs and a pie chart were added in the results section to increase clarity. 

h. Minor edits

Please correct the following:

(h1) Line 147: change “to Sub-study 2” to “in Sub-study 2”

Corrected

(h2) Lines 148 and 149: change “classes” to categories”.

Corrected

(h3) Line 149: the sentence should read “... level categories were defined according to the International Classification of Education …”.

Corrected

(h4) Line 195: change “assistance to the supervisor” to “supervisor’s assistance”.

Corrected

(h5) Line 216: change “…any influence of the ability of the participant…” to “…any influence in the ability of the participant…

Corrected

4. Discussion

Findings are discussed in the context of the published literature and the references cited are relevant to the study. The data supports the conclusions of the work.

The discussion is long. The paragraphs related to the impact that the serology tests can have in fighting the current epidemic, regulations imposed by FDA, CDC and European health authorities and the need to assess the practicability of self-administered assays before promoting their use, belongs to the introduction of the paper.

The entire paragraph and the following has been moved to the introduction (lines 83 – 113)

g. Minor edits

(g1) Line 262: delete “These results are currently…” and add the reference after the previous sentence.

Corrected

---

## [Decision Letter · Decision Letter 1]

15 Dec 2020

PONE-D-20-21033R1

Evaluation of the Practicability of a Finger-Stick Whole-Blood SARS-Cov-2 Self-Test Adapted for the General Population.

PLOS ONE

Dear Dr. Prazuck,

Thank you for submitting your manuscript to PLOS ONE. After careful consideration, we feel that it has merit but does not fully meet PLOS ONE’s publication criteria as it currently stands. Therefore, we invite you to submit a revised version of the manuscript that addresses the points raised during the review process.

Your manuscript was seen by the same reviewers who had reviewed the earlier version. While you claim that the grammatical and editorial errors have been fixed the reviewers remain concerned that the manuscript is still not ready for publication. I urge you to make serious efforts to fix the outstanding issues so that a decision can be made regarding the suitability of your paper for publication.

We look forward to receiving your revised manuscript.

Kind regards,

Sanjai Kumar

Academic Editor

PLOS ONE

Reviewers' comments:

Reviewer's Responses to Questions

**Comments to the Author**

1. If the authors have adequately addressed your comments raised in a previous round of review and you feel that this manuscript is now acceptable for publication, you may indicate that here to bypass the “Comments to the Author” section, enter your conflict of interest statement in the “Confidential to Editor” section, and submit your "Accept" recommendation.

Reviewer #1: (No Response)

Reviewer #2: (No Response)

2. Is the manuscript technically sound, and do the data support the conclusions?

Reviewer #1: Yes

Reviewer #2: Yes

3. Has the statistical analysis been performed appropriately and rigorously? 

Reviewer #1: Yes

Reviewer #2: No

4. Have the authors made all data underlying the findings in their manuscript fully available?

Reviewer #1: Yes

Reviewer #2: Yes

5. Is the manuscript presented in an intelligible fashion and written in standard English?

Reviewer #1: No

Reviewer #2: No

6. Review Comments to the Author

Reviewer #1: Authors state that the revised version of the manuscript was proofread by a native English speaker, but the paper still contains multiple grammatical errors and missing/extra words.

I suggest authors check the references: after several paragraphs were moved from discussion to introduction, the references are out of order. In addition, the reference to the FIND foundation website is not formatted as a reference (line 79).

Methods and Materials:

Substudy 1 (lines 157 – 158): It is not clear from the sentence whether the volunteers read the test’s Instructions for Use, watched an instructional video, or both.

Results: Authors indicate that 142 volunteers were selected to participate, and 140 of those filled in the satisfaction questionnaire. All sections of Table 4 therefore include 140 participants, with the exception of the Age section, which lists 142 participants, and the N in the Substudy 1 table heading. Please revise the table so that the numbers are consistent.

Substudy 1 (lines 223 and 232): the number of participants who filled in the questionnaire is 141 in this section of the text, which contradicts the previous paragraphs and Table 4, and it appears that this number was used in all percentage calculations in these two paragraphs. Please revise the numbers so that the number of participants who filled in the questionnaire is consistent throughout the manuscript.

Substudy 2 (line 260): the description of the study specifies that the number of participants who filled in the questionnaire is 141. Please revise the numbers so that the number of participants who filled in the questionnaire is consistent throughout the manuscript.

Discussion (lines 303 – 306): The order of these two sentences seems to suggest that the first sentence (“Nevertheless, we observed…”) supports the conclusion in the second sentence (“This strongly suggests that the execution...”). Since it doesn’t appear that the authors intended for these two sentences to be logically linked, I suggest revising the passage or at least changing the order of the sentences.

Although seropositivity typically occurs later in the course of disease, as authors correctly noted, based on the currently available data it is not clear whether the antibody positivity confers immunity. In addition, infection status cannot be inferred from a result of a serological test. To this end, the COVID-PRESTO test IFU will recommend further molecular testing to diagnose acute infection; however, this instruction seems to only apply in case of a positive IgM test. It has been shown that for SARS-CoV-2 IgM and IgG seroconversion times are not significantly different, and some studies reported that up to 60% of patients were positive for IgG within the first seven days since onset of symptoms (PMID: 32964627). Therefore, I would suggest not excluding IgG positive patients from this recommendation in cases where acute infection is suspected.

Reviewer #2: MERITS

The manuscript has improved substantially after the authors edited the text, added graphs and addressed reviewers’ comments. However, there are still several changes/edits that need to be included. I mentioned several examples (see minor edits), but I strongly encourage the authors to consult a scientific editor.

MAJOR ISSUES

1. Table 4: Although the authors explained that there is no difference between N(%) and % (under characteristics) and that they state that the table was updated, Table 4 as it stands in the revision, looks identical to the one submitted in the original manuscript. In addition, no changes to the totals or proportions were made. Please update Table 4 accordingly.

2. The comparisons conducted to determine whether there was a difference in participants’ comprehension when instructions were delivered in writing or by video, are not clear. Addition of a table (including groups and results) may facilitate readers’ understanding.

3. Figure 3: what does the asterisk over the column indicate? Please add this information in the figure legend.

MINOR EDITS

4. Line 117: delete easy-to-use device

5. Line 118: replace “In summary” by “Briefly”

6. Line 136: delete “when” and capitalize “A”.

7. Line 137: replace “followed by assay buffer added to the buffer well” with “followed by addition of assay buffer to the buffer well”.

8. Line 138: delete the “s” in “migrates” (complexes is plural).

9. Line 155: Add a stop after “(if any)”.

10. Line 158: delete “the” before study. Start sentence with “Study volunteers…”

11. Lines 168-169: the sentence needs further editing for clarity. Replace “information” by “instructions”.

12. Line 170: sentence should read “The instructions WERE as follows…”

13. Line 177: delete “on the test”.

14. Line 184: change title to “Reading and interpretation of results”.

15. Line 188: replace “their correctness” to “accuracy”.

16. Please revise reference # 2. The citation is incomplete.

17. Table 2, question 3. I suggested the authors to change this question to “Did you manage to collect the recommended amount of blood?” In their response, they state that they made the requested modification. However, Table 2 as it stands in the revision, does not reflect the change.

18. Table 2, question 6. I suggested the authors to change this question to “Did you obtain a valid result (at least one band)?” In their response, they state that they made the requested modification. However, Table 2 as it stands in the revision, does not reflect the change.

19. Table 3. I suggested the authors to change the title and table heading to “Supervisor’s Feedback”. In their response, they state that they made the requested modification. However, Table 2 as it stands in the revision, does not reflect the change.

20. Line 199: delete the word “their”.

21. Line 205: add a comma after “among those”.

22. Line 208: add “:” after defined (…categories were defined:…).

23. Line 211: number “6” is missing.

24. Line 220: replace “on the 221 comprehension of the instructions notice” with “ on comprehension’s instruction”.

25. Line 222: add a comma after “However”.

26. Line 223: add a comma after “Indeed”.

27. Line 258: delete the word “classes” after age and education.

28. Line 273: same as in line 258.

7. PLOS authors have the option to publish the peer review history of their article (what does this mean?). If published, this will include your full peer review and any attached files.

Reviewer #1: No

Reviewer #2: No

---

## [Author Response · Author response to Decision Letter 1]

4 Jan 2021

All comments are adressed in the rebuttal letter.

---

## [Editor Report · Decision Letter 2]

11 Jan 2021

Evaluation of the Practicability of a Finger-Stick Whole-Blood SARS-Cov-2 Self-Test Adapted for the General Population.

PONE-D-20-21033R2

Dear Dr. Prazuck,

We’re pleased to inform you that your manuscript has been judged scientifically suitable for publication and will be formally accepted for publication once it meets all outstanding technical requirements.

Kind regards,

Sanjai Kumar

Academic Editor

PLOS ONE
---

## [Editor Report · Acceptance letter]

20 Jan 2021

PONE-D-20-21033R2 

Evaluation of the Practicability of a Finger-Stick Whole-Blood SARS-Cov-2 Self-Test Adapted for the General Population. 

Dear Dr. Prazuck:

I'm pleased to inform you that your manuscript has been deemed suitable for publication in PLOS ONE. Congratulations! Your manuscript is now with our production department. 

Kind regards, 

on behalf of

Dr. Sanjai Kumar 

Academic Editor

PLOS ONE